# Identification of the *AKCDPK* gene family and *AkCDPK15* functional analysis under drought and salt stress

**Penghua Gao, Ying Zou, Min Yang, Lifang Li, Ying Qi, Jianwei Guo, Yongteng Zhao, Jiani Liu, Jianrong Zhao, Feiyan Huang\*, Lei Yu\***

College of Agronomy, Yunnan Urban Agricultural Engineering and Technological Research Center, Yunnan Provincial Science and Technology Department, Kunming University, Kunming, China

\* yulei0425@163.com (LY); 125593879@qq.com (FH)

## Abstract

Konjac is one of the important economic crops for poverty alleviation in mountainous areas of Yunnan Province, China. However, there are always various biotic and abiotic stress during its growth, leading to production reduction and quality decline. Calcium-dependent protein kinases (CDPKs) are an important class of genes involved in calcium ion signal transmission within plant tissue cells, yet their presence and functions in konjac remain unexplored. This study aimed to identify the members of the *AkCDPK* gene family in the *Amorphophallus konjac* genome and understand their evolution and responses to various stresses. A total of 29 *AkCDPK* genes were identified and categorized into four subgroups that unevenly distributed across 12 chromosomes. Most *AkCDPK* have undergone purifying selection during evolution. Cis-acting element analysis revealed that several *AkCDPK* are involved in phytohormone induction, defence, stress response, and plant development. Expression analysis indicated tissue specificity, and responses to salt, drought, and *Pectobacterium carotovorum* subsp. *carotovorum* stress. *AkCDPK15,* encoding 582 amino acids, was cloned. *AkCDPK15* was mainly localised on the cell membrane, and overexpression in tobacco revealed that it can positively regulate the tolerance of transgenic tobacco strains to salt and drought stress. These findings provide a theoretical foundation for future research on the function of the *CDPK* gene family in *A. konjac*, potentially aiding in the development of stress-resistant konjac varieties.

## Background

The complex growth environment of plants means that they may be subjected to various biotic and abiotic stresses during their lifetimes. Therefore, plants adapt to various harsh environments by regulating their internal environment. These defence responses include a burst of reactive oxygen species (ROS), $Ca^{2+}$-mediated

**Data availability statement:** All relevant data are within the paper and its Supporting Information files.

**Funding:** The funder of Yunnan Education Department Research Project (grant no. 2022J0644, 2023J0827) play a role in study design and decision to publish; The funder of applied Basic Research Foundation of Yunnan Province (grant no. 202301AU070136, 202301AT070055), Kunming University Talent Program (grant no. YJL23010, YJL23005, YJL23007) play a role in data collection and analysis; The funder of Yunnan Key Laboratory of Konjac Biology, Yunnan Province Yu Lei Expert Grassroots Research Workstation, Yunnan Provincial Science and Technology Department (grant no. 202201AU070043, 202101BA070001-174), Yunnan Provincial Science and Technology Department (grant no. 202449CE340009) play a role decision to publish; The funder of College Student Innovation Training Program Project (grant no.S202411393051, S202411393042) had no role in study design, data collection and analysis, decision to publish, or preparation of the manuscript.

**Competing interests:** The authors have declared that no competing interests exist.

**Abbreviations:** ABA, abscisic acid; ATP, adenosine triphosphate; BLASTP, basic local alignment search tool for proteins; CaMs, calmodulins; CAT, catalase; CDPKs, calcium-dependent protein kinases; CBD, calcium-binding domain; FPKM, fragments per kilobase of transcript per million mapped reads; GFP, green fluorescent protein; GRAVY, grand average of hydropathy; GTP, guanosine triphosphate; GSDS, gene structure display server; iTOL, Interactive Tree Of Life; JA, jasmonic acid; MS, Murashige and Skoog; MW, molecular weight; OE, overexpression; ORF, open reading frame; Pcc, *Pectobacterium carotovorum* subsp. *Carotovorum*; Pfam, protein families database; PI, isoelectric point; PKD, protein kinase domain; POD, peroxidase; qRT-PCR, quantitative real-time PCR; ROS, reactive oxygen species; SA, salicylic acid; SOD, superoxide dismutase; VNTD, N-terminal variable domain; WT, wild type.

signalling pathways, and mitogen-activated protein kinase cascade reactions, etc. [1–3]. Calcium-dependent protein kinases (CDPKs) can direct $Ca^{2+}$ signals to phosphorylation pathways, serving as a sensor and responder to $Ca^{2+}$ when plants are under stress [4–7].

CDPKs are serine/threonine protein kinases widely distributed in plants [8]. The proteins encoded by *CDPK* usually contain four domains: calcium-binding (CBD), N-terminal variable (VNTD), Ser/Thr protein kinase (PKD), and self-inhibitory junction (JD) [9]. The CBD contains EF-hand-shaped domains that bind $Ca^{2+}$. Among the four domains, the PKD is highly conserved and contains Ser/Thr phosphorylation sites, which are involved in activating kinase activity. The VNTD region of most *CDPK*-encoded proteins contain myristoyl or palmitoyl sites; the JD, usually located near the PKD, inhibits kinase activity and prevents the occurrence of regulatory pathways by binding to pseudo substrates [7,10]. When plants perceive unfavourable environmental conditions, specific $Ca^{2+}$ signals are generated within the cells. $Ca^{2+}$ directly binds to the EF-hand structure, relieves the self-inhibitory effect of kinases, realises kinase substrate phosphorylation, and activates kinase activity. In addition, the PKD can bind adenosine triphosphate (ATP) or guanosine triphosphate (GTP) and transfer γ-phosphate groups to receptor hydroxyl residues, activate substrates, and induce various physiological reactions in plants [11,12].

Recent studies have indicated that *CDPKs* play a role in plant growth and stress response. When rice (*Oryza sativa*) is subjected to water deficiency, *OsCDPK5* enhances its resistance to drought by increasing its photosynthetic activity [13]. Potato (*Solanum tuberosum*) *CDPKs*, such as *StCDPK28*, have similar functions in response to water deficiency [14]. *OsCPK12* positively regulates rice productivity by prolonging its growth period [15]. *SpCPK33* enhances cold resistance in tomatoes by reducing ROS accumulation and regulating the expression of stress-related genes [16]. In wheat (*Triticum aestivum*), *TaCDPK27* negatively regulates wheat resistance to powdery mildew (*Blumeria graminis* f. sp. *tritici*) by participating in the regulation of ROS production, antioxidant enzyme activity, and programmed cell death processes [17]. And the positive regulatory gene *TaCDPK7* in wheat is involved in tolerance to *Puccinia striiformis* f. sp. *tritici* by regulating the content of hydrogen peroxide and the expression of related defence genes [18].

Konjac is an economically important crop in China. *Amorphophallus konjac* is the most-commonly cultivated konjac species in agriculture. However, *A. konjac* is often threatened by drought, salt stress, and soft rot during cultivation, which hinders the development of the konjac industry. As the *CDPK* gene family acts as receptors for calcium ions, playing important roles in regulating plant growth and development, hormone responses, and a variety of biotic and abiotic stresses responses, we suspected that it plays a key role in the water deficit, salt, and disease resistance of *A. konjac*. At present, the complete genome sequencing of *A. konjac* has laid the foundation for exploring the expression, evolution, and functional characteristics of *CDPK* in *A. konjac* at the whole genome level [19,20]. Consequently, to explore the function and genetic evolution of *AkCDPK*, bioinformatics were used to identify and analyse the *AkCDPK* gene family members from the whole genome of *A. konjac*.

Quantitative real-time PCR (qRT-PCR) was employed to analyse the expression levels of the *AkCDPK* gene family under *Pectobacterium carotovorum* subsp. *carotovorum* (Pcc) stress and abiotic stressors, such as mannitol and salt. The function of *AkCDPK15* was studied by overexpression in tobacco (*Nicotiana benthamiana*). This study provides a theoretical foundation for future research on *AkCDPK* functions in *A. konjac*, potentially aiding in the development of stress-resistant konjac varieties.

## Methods

### Plant materials and treatments

*A. konjac* was supplied by the Konjac Genetic Research Center (Kunming University, Kunming, Yunnan, China). Two-month-old *A. konjac* seedlings were grown in a greenhouse at 28±2 °C, 6000 lx, a relative humidity of 75%, and 16 h/8 h (day/night). Nine *A. konjac* seedlings with the same growth rate were collected from each treatment in triplicates.

For abiotic stress, two-month-old *A. konjac* seedlings were treated with 200 mM NaCl and 100 mM mannitol for 0, 24, and 48 h to induce salt and drought stress.

For biotic stress, two-month-old *A. konjac* seedlings were inoculated with either sterile water or Pcc bacterial suspension (100 μL; $10^8$ cfu/mL) [21]. The sites injected with sterile water served as controls. An inoculation site was selected for each plant. The cultivation conditions for the plants were kept unchanged. The samples were collected at 0, 24, 48, and 72 h.

### Identification of CDPKs in *A. konjac* by bioinformatic analysis

The *A. konjac* genome database was downloaded from https://doi.org/10.6084/m9.figshare.15169578. *Arabidopsis CDPK* protein sequences were downloaded from https://www.arabidopsis.org/ (S1 Table). *O. sativa CDPK* sequences were downloaded from http://rice.uga.edu/pub/data/Eukaryotic_Projects/o_sativa/annotation_dbs/pseudomolecules/version_7.0/all.dir/ (S1 Table). *Arabidopsis* and rice CDPK amino acid sequences were employed to construct a hidden Markov model to search the HMMER 3.0 (http://hmmer.janelia.org/) software [22] to identify all potential *AkCDPK* family numbers of *A. konjac*. The candidate reference sequences were compared using the BLASTP (version: ncbi last v2.10.1+) [23] with an e-value of 1e-20. The sequences containing the PF00069 and PF13499 domains were then annotated as *AkCDPK* gene family numbers by pfamscan (version: v1.6) and Pfam A (version: v33.1) [24,25] (S1 Table).

### Sequence analysis of *AkCDPK*

The bioinformatics analysis of *AkCDPK* were predicted using the ExPASY PROTPARAM tool (http://web.expasy.org/protparam/). MEME software (version: v5.0.5) [26] was used to analyse the similarity and diversity of protein motifs, and conserved motifs of AkCDPK family numbers. MG2C (http://mg2c.iask.in/mg2c_v2.1/) [27] was employed to draw the chromosome physical location map of the *AkCDPK* family numbers. The transmembrane domains, signal peptides, and subcellular localisation were predicted by DeepTMHMM (version 1.0.8), SignalP (version: v5.0b), and WOLFPSORT (https://wolfpsort.hgc.jp/), respectively. The transcription factor (TF) binding sites in each *AkCDPK* was predicted using the PlantCARE software (http://bioinformatics.psb.ugent.be/webtools/plantcare/html/) [28].

MAFFT (version: v7.427) [29] was employed to perform multiple alignments of CDPK sequences from *A. konjac* (AkCDPK), *Arabidopsis thalliana* (AtCDPK), and *Oriza sativa* (OrCDPK). MEGA (MEGA10) software was used to construct phylogenetic tree (neighbour-joining (NJ) method, 1000 bootstrap) using the p-distance model [30]. The NJ tree was then annotated by the interactive Tree of Life v6 software (https://itol.embl.de/) [31]. The AkCDPK protein multiple sequence alignment was performed using Jalview software [32] to further elucidate the characteristics of the *AkCDPK* family. And the GSDS tool (http://gsds.cbi.pku.edu.cn/) [33] was employed to display the gene structure.

## Selection pressure analysis of the *AkCDPK* family

The non-synonymous substitution rate (Ka)/ synonymous substitution rate (Ks) ratios of paralogous *AkCDPK* gene pairs were calculated using KaKs_Calculator (version: 2.0) software to determine whether *AkCDPKs* had selection pressure [34].

## Collinearity analysis of *AkCDPK*

MCScanX software was employed to conduct the collinearity analysis of *AkCDPK* [35].

## Expression profiles of *AkCDPK*

The expression profiles of *AkCDPK* in the roots, corms, petioles, and leaves were obtained from the NCBI (S2 Table). The expression levels under salt stress, drought stress and Pcc stress treatments were determined using qRT-PCR.

 *A. konjac* seedlings were cultivated at 27±2 °C for one months. Total RNA was extracted from the all tested samples using the Trelief ® Hi-Pure Plant RNA Plus Kit (Tsingke Biotechnology Co., Ltd., Beijing, China), the total RNA was revised to cDNA using SynScript® III cDNA Synthesis Mix (Tsingke). Followed by the 1.1×EasyQ SYBR qPCR Mix (Low ROX Premixed) EasyQ (Tsingke) was employed to conduct qRT-PCR. All experimental protocols were based on the manufacturer's instructions. Each reaction carried out with three technical replicates. The $2^{-(\Delta\Delta Ct)}$ method [36] was used to calculate relative expression levels of the tested *AkCDPK* genes. The qRT-PCR primers were shown in S3 Table.

### *AkCDPK15* subcellular localisation and phosphorylation site analyses

The sequence 35S::AkCDPK15::GLOsGFP was generated by amplifying the coding sequence of *AkCDPK15* without the terminal codon from *A. konjac* cDNA, using the primers pBWA(V)HS-AkCDPK15-GLOsGFP-F (AGAGAACAC GGGGGACTTTGCAACATGGGCAACACATGCCGC) and pBWA(V)HS-AkCDPK15-GLOsGFP-R (GTACTGAAGA CAGAGCTAGTTACATTAGGATGCCAATGGAGAACCTCTCATG). The PCR fragment was cloned into the pBWA(V) HS-GLOsGFP vector using the MonClone™ Single Assembly Cloning Mix (Mona [Suzhou] Biotechnology Co., Ltd., Suzhou, China). The constructed expression vector pBWA (V) HS-AkCDPK15 Glosgfp was added to GV3101 for cultivation. After expansion, the bacterial cells were collected and resuspended in a 10 mM $MgCL_2$ (containing 120 µM AS) suspension (OD = 0.6–0.8). The lower epidermis of tobacco was then injected with 1 ml of the above suspension. After injection, the cells were cultured under dark conditions for two days. Labelled leaves were removed, and glass slides were observed under a laser confocal microscope and photographed.

 Western blot analysis was performed on transgenic tobacco leaves using transient expression of the pBWA (V) HS-AC DPK15-Glosgfp vector (S1 Fig). The target band was cut and enzymatically hydrolysed using trypsin. The hydrolysed peptide segments were desalted and detected using high-performance liquid mass spectrometry. Finally, the mass spectrometry data were analysed using PD software with the parameter phospho for variable modification, fixed modification for carbammidomethyl (C), and trypsin/P for enzyme digestion, with the first-level mass spectrometry matching tolerance was 3 ppm; the secondary mass spectrometry matching was 20 ppm.

### *AkCDPK15* gene function analysis

To explore the function of the *AkCDPK15* gene, a plate germination experiment was conducted using transgenic lines obtained as follows.

 The sequence 35S::*AkCDPK15*::GFP was generated by amplifying the coding sequence without the terminal codon of *AkCDPK15* from *A. konjac* cDNA using primers pCAMBIA-*AkCDPK15*-F (GAGGACAGGGTACCCGGGGATCCAT-GGGCAACACATGCCGC) and pCAMBIA-*AkCDPK15*-R (CTAGTGTCGACTCTAGAGGATCCTTAGGATGCCAATG-GAGAACCTCTCATG). The PCR fragment was cloned into the *pCAMBIA-1300* vector using the MonClone™ Single

Assembly Cloning Mix (Mona [Suzhou] Biotechnology Co., Ltd., Suzhou, China). Followed by transforming the successfully constructed *pCAMBIA-AkCDPK15* plasmid into *Agrobacterium tumefaciens* strain GV3101. This was followed by the transformation *of A. tumefaciens* carrying *the pCAMBIA-AkCDPK15* vector into *N. benthamiana* leaves and overexpression of *AkCDPK15* via the leaf disc transformation method. The obtained transgenic strains were amplified using specific primers, pCAMBIA-AkCDPK15-FA (TTCATTTGGAGAGAACACGGGGGAC) and pCAMBIA-AkCDPK15-RA (GATTGAGAGGAAGGACAACCAA) (S2 Fig). The overexpressing *AkCDPK15* representative lines were used for further analyses.

First, transgenic (OE1, OE2, OE3) and wild-type (WT) strains were cultured on a flat plate (130 mm*130 mm) containing ½ Murashige & Skoog (MS) medium supplemented with 200 mM NaCl and 100 mM mannitol, and a statistical analysis on the root length of *AkCDPK15* was conducted after 14 d culture (n = 10).

Next, one-month-old *AkCDPK15*-overexpressing and WT tobacco strains were subjected to water deficit treatment. Twenty tobacco plants for both *AkCDPK15* transgenic strain and WT strain were subjected to drought stress conditions. For the drought stress treatment, water was withheld from the *AkCDPK15* transgenic and WT strains for two weeks, followed by regular watering for two days. Malondialdehyde (MDA), $H_2O_2$, proline, soluble sugar accumulation, and antioxidant enzyme activity were then measured.

For $H_2O_2$ accumulation assessment, detection kits for MDA content, soluble sugar content, proline content, and superoxide dismutase (SOD), catalase (CAT), and peroxidase (POD) enzyme activities (Beijing Solarbio Technology Co., Ltd., Beijing, China) were used with spectrophotometry.

## Results

### Identification of AkCDPK family members in *A. konjac*

A total of 29 putative CDPK proteins were identified in the *A. konjac* genome and named AkCDPK1–AkCDPK29. The full length of the 29 AkCDPK proteins varied from 497 (AkCDPK1) to 616 (AkCDPK29) amino acids (aa), with coding sequence (CDS) lengths ranging from 1494 to 1851 bp. The MW varied from 51.80 (AkCDPK3/11) to 66.94 KDa (AkCDPK29), whereas the theoretical isoelectric points (pI) varied from 5.16 (AkCDPK2) to 9.08 (AkCDPK21). Among the family members, most genes encoded unstable protein. The grand average of hydropathy (GRAVY) varied from −0.59 to −0.186, indicating that all AkCDPK are hydrophilic. Most AkCDPK contained both the predicted N-myristoylation and S-palmitoylation sites, whereas AkCDPK3 contained only one predicted N-myristoylation site. AkCDPK1, 4, and 19 were predicted to contain no N-myristoylation or S-palmitoylation sites. All AkCDPK harboured protein kinase and EF-hand domains (Table 1).

### Phylogenetic, gene structure, and chromosomal distribution of *AkCDPK*

To understand the phylogenetic relationships among the CDPKs in *A. konjac*, CDPK sequences of three species, *A. konjac*, *A. thaliana*, and *O. sativa*, were employed to construct NJ tree using MEGA10. The CDPK sequences grouped into four subfamilies, with 7, 5, 12, and 5 members in Groups I, II, III, and IV, respectively (Fig 1).

### Gene structure, motif, and chromosomal location analyses of *CDPK* in *A. konjac*

To understand the possible structural evolution of *AkCDPK*, the diversity of the exon-intron organisation within the *AkCDPK* family was compared. As shown in Fig 2A, all *AkCDPK* members possessed 5–11 introns (3 with 5 introns, 9 with 6 introns, 13 with 7 introns, and 4 with 11 introns). In Group I, members had seven or eight exons. A diverse number of exons were found in Group II: six exons were found in *AkCDPK24*, *AkCDPK26*, and *AkCDPK28*; seven exons were found in *AkCDPK7*, *AkCDPK8*, and *AkCDPK18*; and eight exons were found in *AkCDPK2*, *AkCDPK20*, *AkCDPK5*, *AkCDPK22*, *AkCDPK23*, and *AkCDPK27.* The members in Group III had 7 or 12 exons, of which 7 exons were found in *AkCDPK4* and 12 were found in *AkCDPK14*, *AkCDPK16*, *AkCDPK21*, and *AkCDPK25*. In Group IV, all members had eight exons. These results indicated that *CDPK* with higher homogeneity usually had the same number of exons.

**Table 1. Characteristics of *AKCDPK* in *Amorphophallus konjac*.**

| Gene Name | Gene ID | Chromosome Location | coding sequence length | Amino acid (aa) no. | Molecular weight (Da) | Isoelectric points | GRAVY | No. of EF Hands | N-Acylation Prediction (No.) |
|---|---|---|---|---|---|---|---|---|---|
| AkCDPK1 | evm.model.CTG_17200.13.1_Akon | CTG_17200:263057..267912 | 1494 | 497 | 56254.16 | 5.34 | −0.339 | 2 | 0 |
| AkCDPK2 | evm.model.CTG_17475.4_Akon | CTG_17475:70008..75001 | 1677 | 558 | 62129.62 | 5.9 | −0.328 | 2 | N-Myr (1)–S-Palm (3) |
| AkCDPK3 | evm.model.CTG_18290.6_Akon | CTG_18290:113867..118587 | 1380 | 459 | 51801.14 | 5.47 | −0.345 | 2 | N-Myr (1)–S-Palm (0) |
| AkCDPK4 | evm.model.CTG_20042.12_Akon | CTG_20042:179353..185036 | 1767 | 588 | 65245.22 | 5.35 | −0.332 | 2 | 0 |
| AkCDPK5 | evm.model.CTG_3081.5_Akon | CTG_3081:112268..117036 | 1626 | 541 | 60596.9 | 5.4 | −0.435 | 2 | N-Myr (1)–S-Palm (3) |
| AkCDPK6 | evm.model.CTG_9134.3_Akon | CTG_9134:6417..85398 | 1638 | 545 | 61434.67 | 6.56 | −0.555 | 2 | N-Myr (1)–S-Palm (1) |
| AkCDPK7 | evm.model.HIC_ASM_0.4363_Akon | HIC_ASM_0:209595801..209690834 | 1749 | 582 | 64668.29 | 5.95 | −0.336 | 2 | N-Myr (1)–S-Palm (2) |
| AkCDPK8 | evm.model.HIC_ASM_0.4386_Akon | HIC_ASM_0:210725827..210794238 | 1665 | 554 | 62771.33 | 8.79 | −0.59 | 2 | N-Myr (1)–S-Palm (2) |
| AkCDPK9 | evm.model.HIC_ASM_0.5124_Akon | HIC_ASM_0:237769900..237776387 | 1605 | 534 | 59417.58 | 5.84 | −0.466 | 2 | N-Myr (1)–S-Palm (1) |
| AkCDPK10 | evm.model.HIC_ASM_0.5128_Akon | HIC_ASM_0:237909385..237915869 | 1629 | 542 | 60692.22 | 5.99 | −0.414 | 2 | N-Myr (1)–S-Palm (1) |
| AkCDPK11 | evm.model.HIC_ASM_0.5318_Akon | HIC_ASM_0:243665230..243669940 | 1494 | 497 | 56254.16 | 5.34 | −0.339 | 2 | N-Myr (1)–S-Palm (0) |
| AkCDPK12 | evm.model.HIC_ASM_0.5386_Akon | HIC_ASM_0:245323336..245339406 | 1629 | 542 | 60526.93 | 5.16 | −0.296 | 2 | N-Myr (1)–S-Palm (1) |
| AkCDPK13 | evm.model.HIC_ASM_1.10394.2_Akon | HIC_ASM_1:473829711..473843511 | 1629 | 542 | 60450.88 | 5.19 | −0.292 | 2 | N-Myr (1)–S-Palm (1) |
| AkCDPK14 | evm.model.HIC_ASM_1.1830_Akon | HIC_ASM_1:50282319..50299659 | 1647 | 548 | 62047.69 | 9.08 | −0.514 | 2 | N-Myr (1)–S-Palm (1) |
| AkCDPK15 | evm.model.HIC_ASM_11.3224_Akon | HIC_ASM_11:79853509..79858908 | 1602 | 533 | 60074.41 | 6.02 | −0.505 | 2 | N-Myr (1)–S-Palm (1) |
| AkCDPK16 | evm.model.HIC_ASM_11.5617_evm.model.HIC_ASM_11.5618_Akon | HIC_ASM_11:206243534..206313183 | 1602 | 533 | 60060.38 | 6.02 | −0.506 | 2 | N-Myr (1)–S-Palm (1) |
| AkCDPK17 | evm.model.HIC_ASM_4.1275_Akon | HIC_ASM_4:33212080..33215666 | 1683 | 560 | 63368.4 | 6.25 | −0.483 | 2 | N-Myr (1)–S-Palm (1) |
| AkCDPK18 | evm.model.HIC_ASM_4.2160.1_Akon | HIC_ASM_4:61367073..61413550 | 1647 | 548 | 62028.64 | 9.01 | −0.512 | 2 | N-Myr (1)–S-Palm (2) |
| AkCDPK19 | evm.model.HIC_ASM_4.2288.1_Akon | HIC_ASM_4:64437246..64442101 | 1713 | 570 | 64886.09 | 6.81 | −0.494 | 2 | 0 |
| AkCDPK20 | evm.model.HIC_ASM_4.2664_Akon | HIC_ASM_4:74975034..74980004 | 1602 | 533 | 60057.74 | 5.91 | −0.434 | 2 | N-Myr (1)–S-Palm (3) |
| AkCDPK21 | evm.model.HIC_ASM_5.10001_Akon | HIC_ASM_5:430159214..430172624 | 1713 | 570 | 64818.01 | 6.81 | −0.487 | 2 | N-Myr (1)–S-Palm (2) |
| AkCDPK22 | evm.model.HIC_ASM_5.11603_Akon | HIC_ASM_5:468342621..468347497 | 1851 | 616 | 66940.77 | 4.85 | −0.242 | 2 | N-Myr (1)–S-Palm (2) |
| AkCDPK23 | evm.model.HIC_ASM_5.11616_Akon | HIC_ASM_5:468702494..468707370 | 1380 | 459 | 51801.14 | 5.47 | −0.345 | 2 | N-Myr (1)–S-Palm (2) |
| AkCDPK24 | evm.model.HIC_ASM_5.8866_Akon | HIC_ASM_5:402294432..402317205 | 1563 | 520 | 57603.92 | 5.86 | −0.186 | 2 | N-Myr (1)–S-Palm (4) |
| AkCDPK25 | evm.model.HIC_ASM_5.9948_Akon | HIC_ASM_5:429026695..429046511 | 1599 | 532 | 59535.98 | 6.15 | −0.453 | 2 | N-Myr (1)–S-Palm (2) |
| AkCDPK26 | evm.model.HIC_ASM_9.420_Akon | HIC_ASM_9:11640424..11657737 | 1566 | 521 | 58269.39 | 6.13 | −0.448 | 2 | N-Myr (2)–S-Palm (3) |
| AkCDPK27 | evm.model.HIC_ASM_9.462_Akon | HIC_ASM_9:12209941..12217085 | 1638 | 545 | 60892.25 | 6.12 | −0.457 | 2 | N-Myr (1)–S-Palm (2) |
| AkCDPK28 | evm.model.HIC_ASM_9.721_Akon | HIC_ASM_9:17049898..17063692 | 1632 | 543 | 60748.12 | 6.12 | −0.456 | 2 | N-Myr (2)–S-Palm (3) |
| AkCDPK29 | evm.model.HIC_ASM_9.7625_Akon | HIC_ASM_9:356624016..356745767 | 1677 | 558 | 62129.62 | 5.9 | −0.328 | 2 | N-Myr (1)–S-Palm (1) |

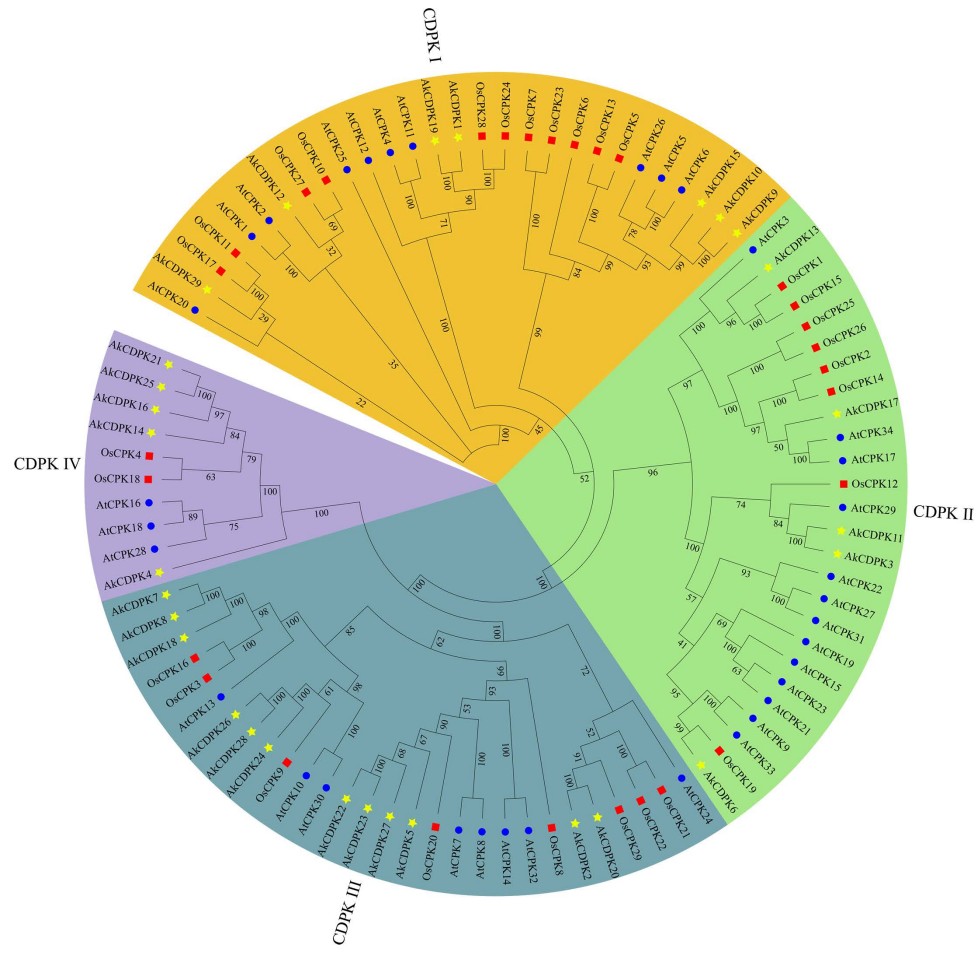

**Fig 1. Phylogenetic relationships of CDPK proteins in *Amorphophallus konjac* (yellow pentagram), *Arabidopsis thaliana* (blue circle), and *Oryza sativa* (red box).** The evolutionary tree was constructed using MEGA10 software with the neighbour-joining method. The numbers next to the branch show 1000 bootstrap replicates, expressed as percentages. The phylogenetic groups of AkCDPK are marked with different colours and legends.

The MEME software was used to further understand the similarity and diversity of AkCDPK protein motifs. Fifteen conserved motifs were predicted in AkCDPK family numbers with similar motif types and sequences. Hence, the 29 AkCDPK proteins had the same conserved motifs and orders. AkCDPK4 contained six motifs. AkCDPK1, AkCDPK19, AkCDPK4, AkCDPK3, AkCDPK11, and AkCDPK16 did not contain a motif 13. Compared to other family members, AkCDPK5, AkC-DPK22, AkCDPK23, AkCDPK27, AkCDPK7, AkCDPK8, AkCDPK18, AkCDPK24, AkCDPK26, and AkCDPK28 contained a unique motif 10 (Fig 2B).

All 29 *AkCDPK* in *A. konjac* were mapped to 12 chromosomes (Fig 3). And there were significant differences in the number of genes on different chromosomes. The largest numbers of genes (*AkCDPK7, AkCDPK8, AkCDPK9, AkC-DPK10, AkCDPK11, AkCDPK12*) were located on chromosome HIC-ASM-0; five genes (*AkCDPK21, AkCDPK22, AkCDPK23, AkCDPK24, AkCDPK25*) were located on chromosome HIC-ASM-5, and chromosomes HIC-ASM-4 and HIC-ASM-9 were found to contain four *AkCDPK* each. Chromosomes HIC-ASM-1 and HIC-ASM-11 contained two *AkC-DPK*. Chromosomes CTG-17200, CTG-17475, CTG-18290, CTG-20042, CTG-3081, and CTG-9134 harboured only one *AtCDPK* each.

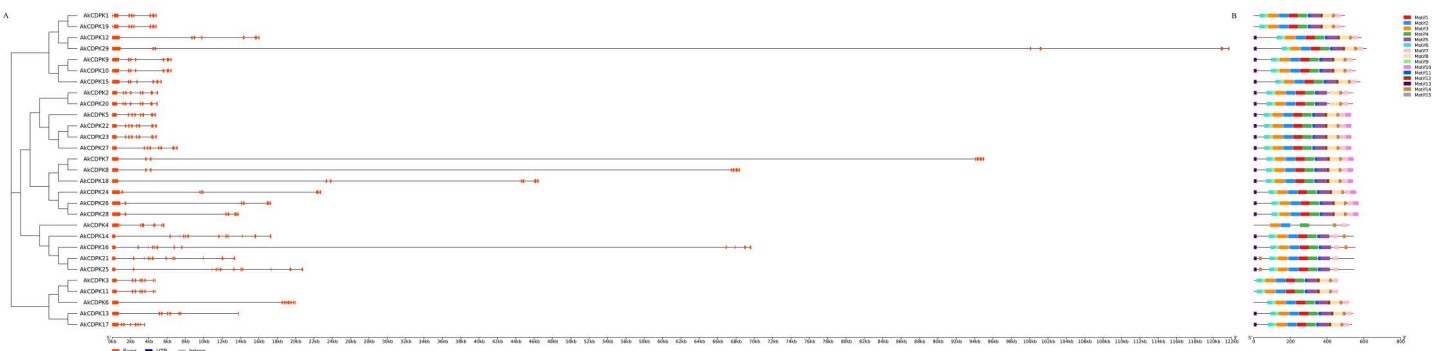

**Fig 2. Phylogenetic relationships, gene structure, and conserved motifs of Ak*CDPK*. (A)** Structure of the *AkCDPK* genes. Orange, black, and blue boxes represent the exons, introns, and untranslated regions, respectively. **(B)** Conserved motifs of AkCDPK proteins. The motifs are indicated by different coloured boxes and their numbers are listed on the right.

To elucidate the replication events of the *AkCDPK* gene family, the MCScanX software was used for collinearity analysis. Two duplicated *CDPK* gene pairs were found in the *A. konjac* genome: one was segmentally duplicated (*AkCDPK14/21*) and the other was tandemly duplicated (*AkCDPK26/28*) (Fig 4A). Furthermore, the collinear correlation among *A. konjac, A. thaliana*, and *O. sativa* was calculated to infer the evolutionary history of *CDPKs*. Two *AkCDPK* showed syntenic relationships with those in *Arabidopsis* and six with those in rice (Fig 4B; S4 Table). Two pairs of syntenic orthologous genes (one-to-one) were identified between *Arabidopsis* and *A. konjac CDPK* genes: *AkCDPK11-AtCPK29* and *AkCDPK21- AtCPK18* (Fig 4B; S4 Table). Between rice and *A. konjac CDPK*, there were two kinds of syntenic orthologous gene pairs: One *A. konjac* gene and multiple rice genes, such as *AkCDPK17-OsCPK2/14/25*, and one *A. konjac* gene vs. one rice gene, such as *AkCDPK9-OsCPK6*, *AkCDPK15-OsCPK13*, *AkCDPK24-OsCPK9*, and *AkCDPK29-OsCPK11* (Fig 4B; S4 Table), indicating that these genes might have been derived from the same ancestor of rice and *A. konjac.* For further evolutionary studies, the number of Ka and Ks were computed to analyse selection pressure (S5 Table). Most orthologous *CDPK* gene pairs had Ka/Ks < 1, indicating those *AkCDPK* numbers underwent strong purifying selective pressure during evolution.

To explore which factors might regulate the level of *AkCDPK* expression, the PlantCARE database was used to predict cis-acting regulatory elements of *AkCDPK* (S6 Table). CAAT-box, activating sequence 1 (as-1), and TATA box are cis-acting elements shared by all *AkCDPK* genes. The remaining cis-acting elements of the genes can be divided into four categories. First, there are multiple phytohormone responsive elements, including salicylic acid response element (TCA-element), gibberellin-responsive element (P-box, TATC-box, and GARE-motif), auxin-responsive element (TGA-box, AuxRR-core), ethylene-responsive element (ERE), ABA response element (ABRE), and MeJA-responsive element (CGTCA-motif and TGACG-motif), suggesting that *AkCDPK* expression might be regulated by multiple phytohormones. Multiple elements were involved in the light response, such as the G-box, Box 4, and TCCC-motif. What's more there are varieties of growth and development elements, including meristem expression elements (CAT-box, CCGTCC motif, and CCGTCC-box), flavonoid biosynthetic gene regulation elements (MBSI), lignin biosynthetic gene regulation elements (AC-I and AC-II), endosperm expression elements (GCN4_motif), seed-specific regulatory elements (RY-element), palisade mesophyll cell elements (HD-Zip 1), and circadian elements (O2-site). In addition, stress-responsive elements were found in *AkCDPK*, for example, the anaerobic induction element (ARE and GC-motif), dehydration-responsive element (DRE core and DRE1), low-temperature-responsive elements, defence and stress-responsive elements (TC-rich repeats), stress-responsive elements (STRE), and wound-responsive elements (WRE3 and WUN-motif) (Fig 5, S6 Table). There were also some transcription factor-binding elements, including the MYB-like sequence, Myb-binding site, and W-box (S6 Table). ABRE, MeJA responsiveness, ARE, and STRE elements were detected in the promoter regions of most *AkCDPK*.

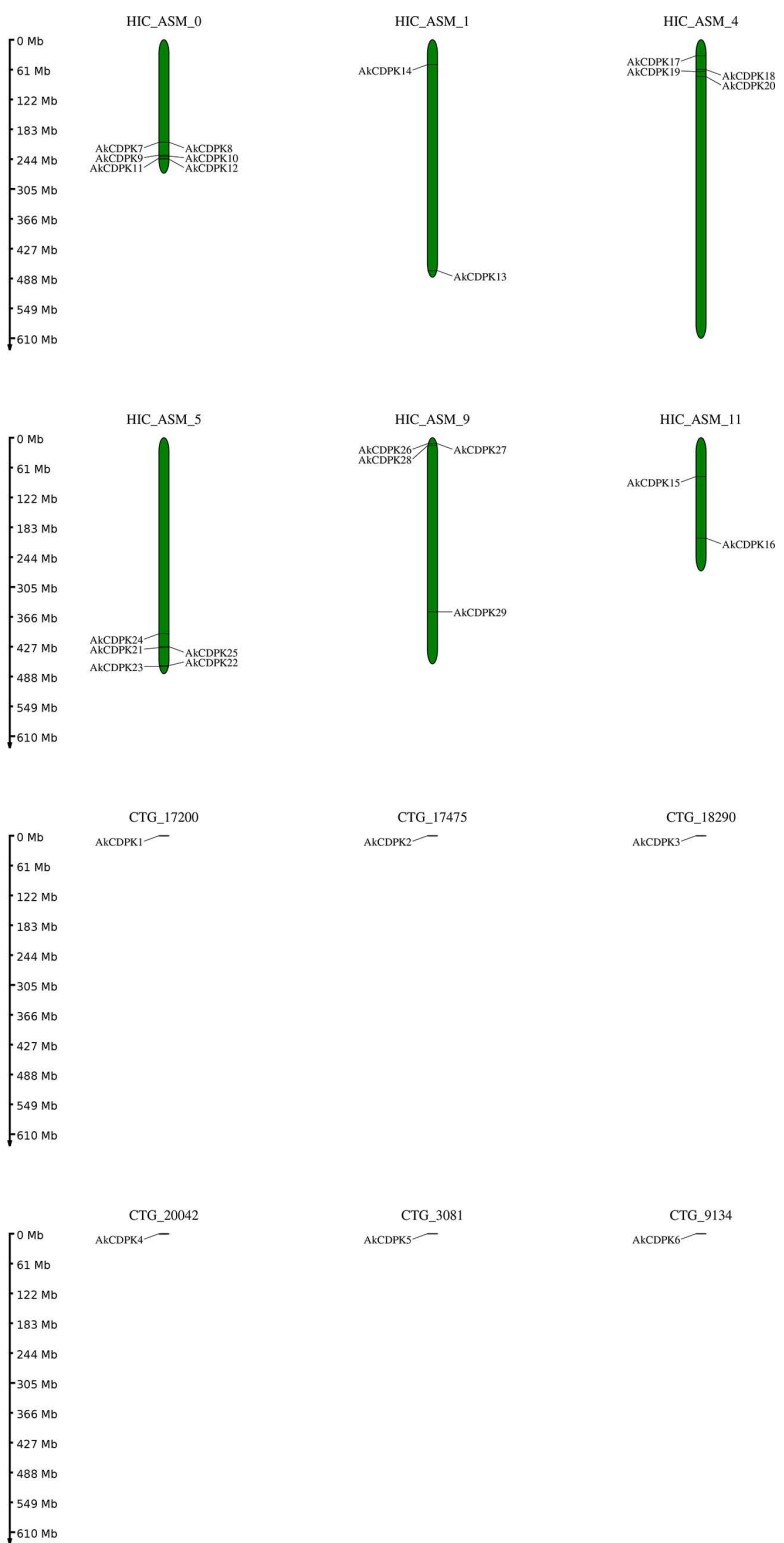

**Fig 3. Physical map of *AkCDPK* chromosome locations.** The vertical axis represents the length of the chromosomes.

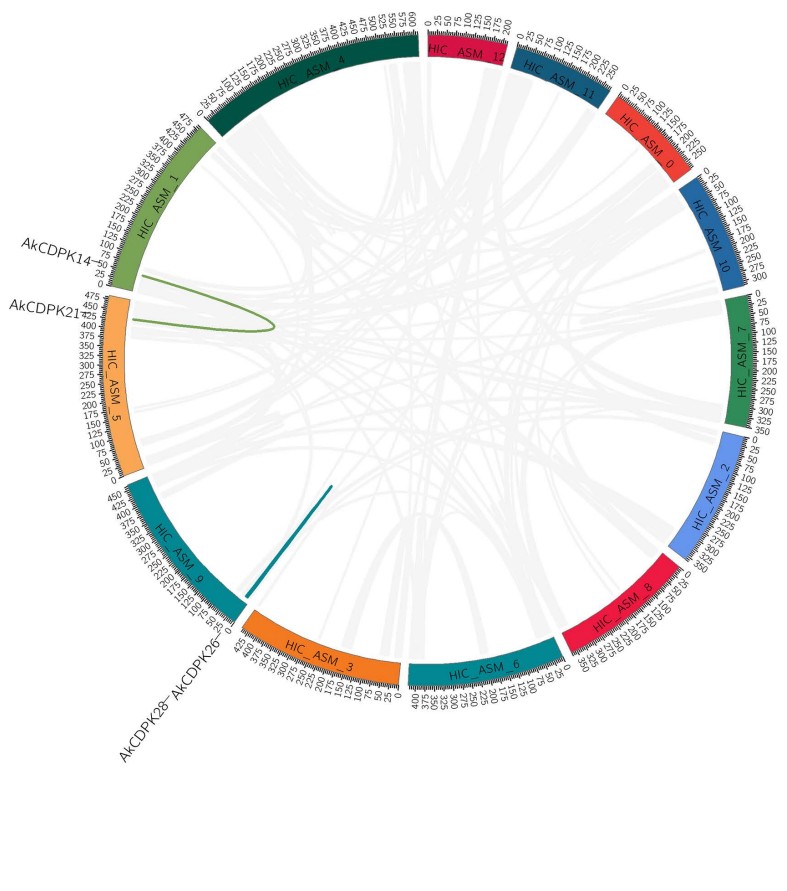

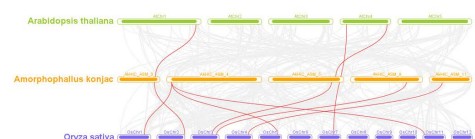

**Fig 4. Synteny analysis of *CDPK* in *Amorphophallus konjac*, *Arabidopsis thaliana*, and *Oryza sativa*. (A)** Schematic representation of the chromosomal distribution and intrachromosomal relationships of *A. konjac CDPK* genes. **(B)** Collinear correlation between *A. konjac* and *A. thaliana,* and *O. sativa*. Grey lines in the background indicate the collinear blocks within *A. konjac* and other plant genomes, whereas the red lines highlight syntenic *AkCDPK* gene pairs.

The above results suggest that *AkCDPK* are not only involved in the growth and development of *A. konjac* but also in its response to various environmental stresses.

### Expression profiles of *AkCDPK* in different tissues

To understand the possible roles of *AkCDPK*, the spatial expression patterns of the 29 *AkCDPK* genes were analysed using RNA-Seq expression data from *A. konjac* recently published by Li et al. [19]. The results showed that *AkCDPK2, 3, 11, 17*, and *20* were not expressed in the root, bulb, leaf, or petiole. The remaining genes had high expression levels in the petiole. Moreover, *AkCDPK8, 9*, 12, and *29* transcript levels in bulbs were significantly higher than those of other genes. Compared with other genes, *AkCDPK6,* and *AkCDPK24* showed higher transcription levels in leaves (Fig 6). These results indicate that *AkCDPK* gene family members play different regulatory roles in the growth and development of *A. konjac.*

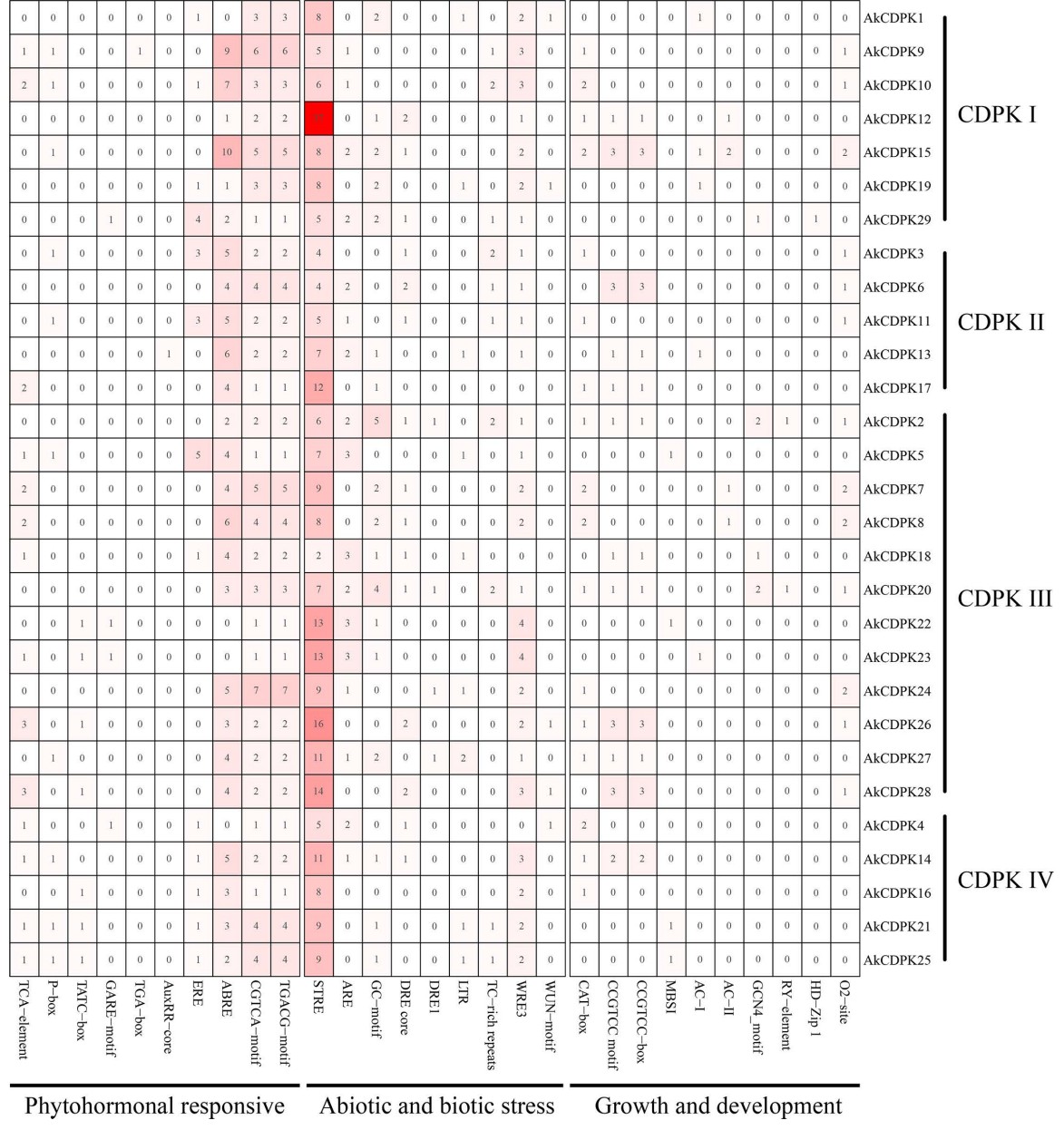

**Fig 5. Cis-acting regulatory elements in the *AkCDPK* promoter region .**

## Expression profiles of *AkCDPK* in response to biotic and abiotic stress

To explore the expression characteristics of *AkCDPK* family members in response to external stress, qRT-PCR was used to analyse the expression patterns of 18 family members under Pcc infection, drought, and salt stress. Most *AkCDPK* family members were induced under Pcc stress, except for *AkCDPK16* and *AkCDPK27*; *AkCDPK22* and *AkCDPK18* were induced at the late stage of Pcc infection (72 h). Most of the genes were induced at the early stage of Pcc infection (24 h) and then downregulated, such as *AkCDPK1/5/6/7/9/12/13/14*. The expression levels of *AkCDPK15*, *AkCDPK21,* and *AkCDPK24* were significantly increased within 72 h of Pcc infection (Fig 7). We found that all 18 *AkCDPK* analysed

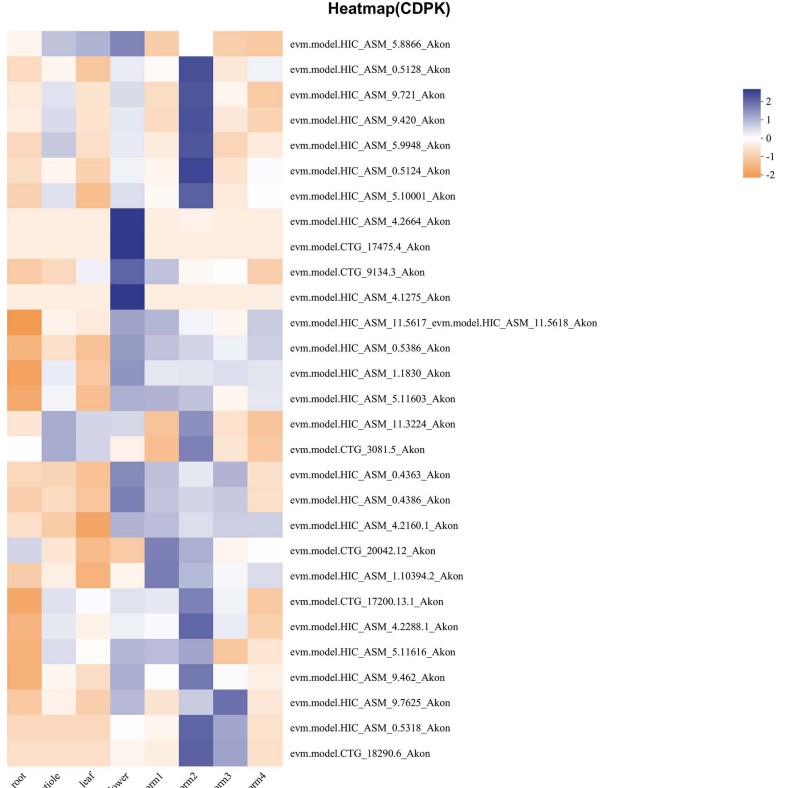

**Fig 6. Expression patterns of *AkCDPK* in different tissues (root, bulb, petiole and leaf).** The heat map shows the relative expression after log10 transformation.

responded to mannitol and salt stress. Among these genes, the expression level of *AkCDPK27* was suppressed with mannitol and salt treatment for 24 h. When *A. konjac* was subjected to mannitol and salt stress for 24 h, the expression levels of *AkCDPK12/13/16* were significantly upregulated, indicating that these three genes may participate in the early stages of osmotic stress. The expression levels of *AkCDPK1/4/5/6/7/9/14/15/18/21/22/24/26/29* were significantly higher than those in the control group within 48 h of mannitol and salt treatment (Fig 8).

## Cloning and characterisation of *AkCDPK15* in *A. konjac*

The expression profile of *A. konjac CDPK* family genes suggested that they play an important role in response to different biotic and abiotic stresses. To further understand their function, a Group I member (*AkCDPK15*) was selected for further study. An open reading frame (ORF) of *AkCDPK15*, which was 1749 bp long and encoded 582 amino acids, was obtained from *A. konjac*. The sequence alignment of *AtCDPK5/26* revealed a high homology with *AkCDPK15*. To characterise the subcellular localisation of AkCDPK15, pBWA(V)HS-AkCDPK15-GLOsGFP fusion protein was transformed into tobacco protoplasts. The result showed that AkCDPK15 was mainly localised in the cell membrane region, with a small amount expressed in the nucleus (Fig 9), indicating that AkCDPK15 mainly functioned at these two sites.

Because the function of *CDPK* genes is often accompanied by protein phosphorylation, we identified the phosphorylation sites of AkCDPK15. The results showed that AkCDPK15 contained eight phosphorylation sites (S87 [88.5], S92 [88.5], T94 [88.5], S359 [77.3], S436 [96.2], S453 [99.7], S508 [83.9], and T526 [90]), and the probability of all phosphorylisable amino acid sites occurring was above 75% (S3 Fig, S7 Table).

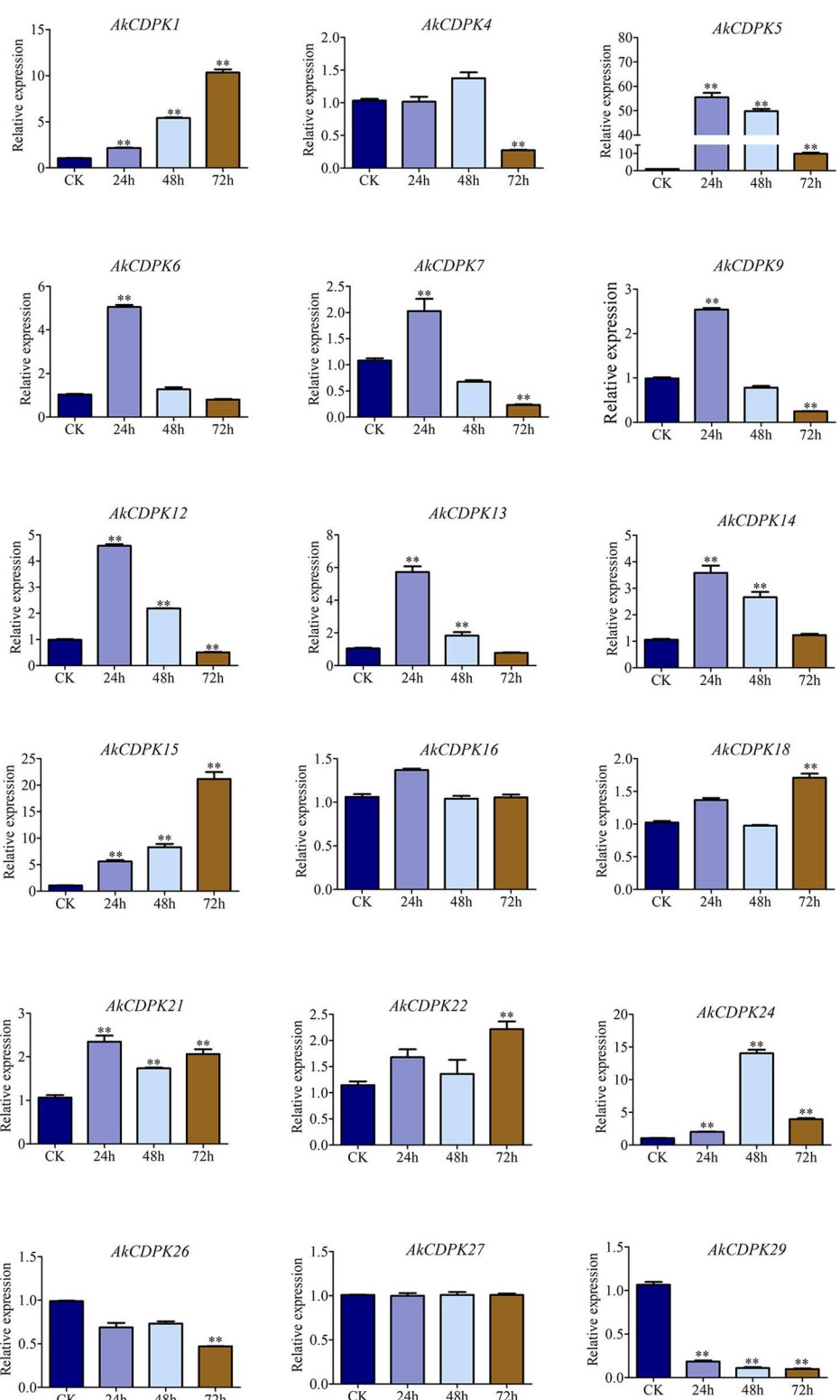

**Fig 7. Expression profiles of *AkCDPK* during *Pectobacterium carotovorum* subsp. *carotovorum* (Pcc) infection.**

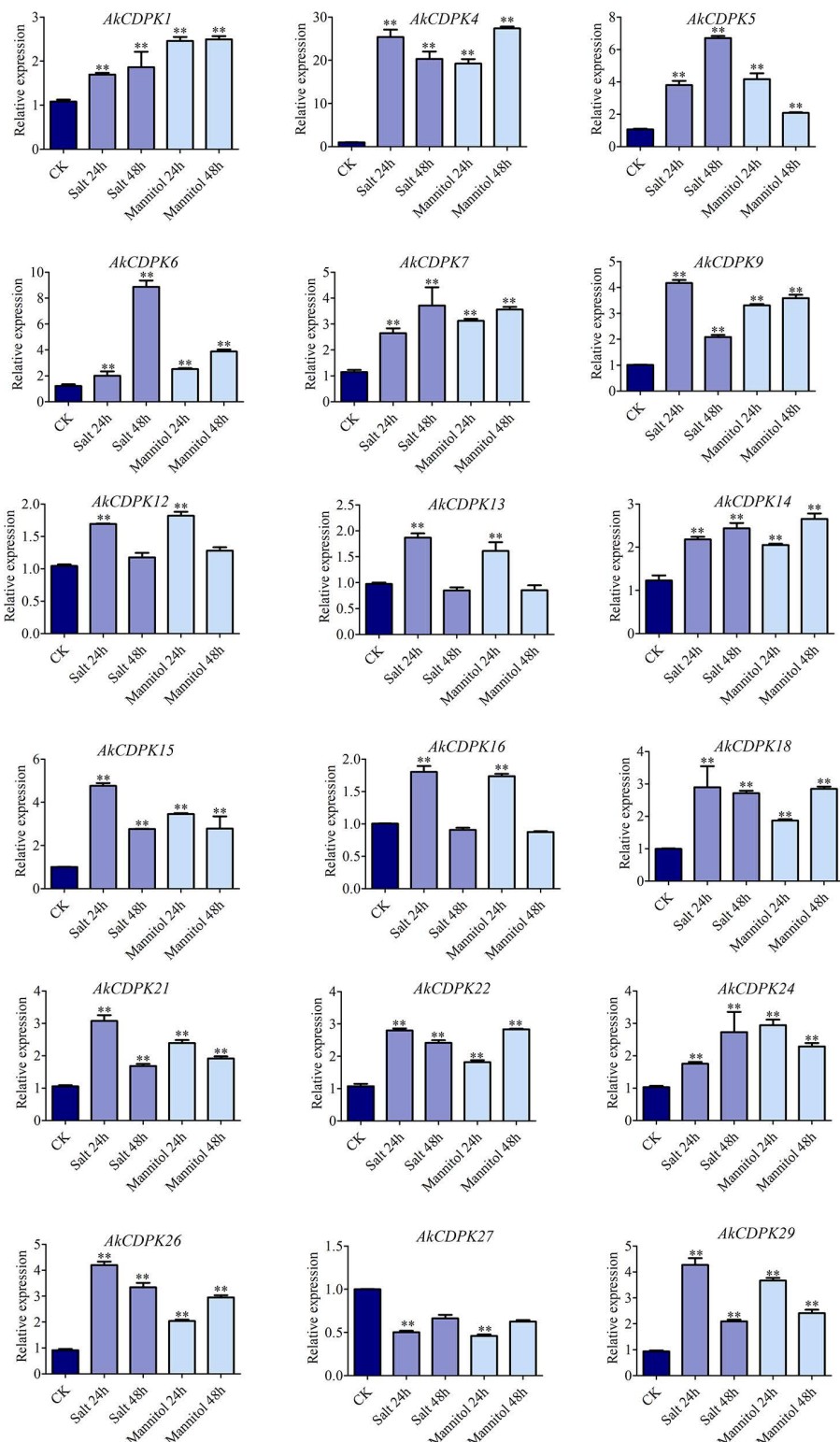

**Fig 8. Expression profiles of *AkCDPK* under salt and mannitol stress.**

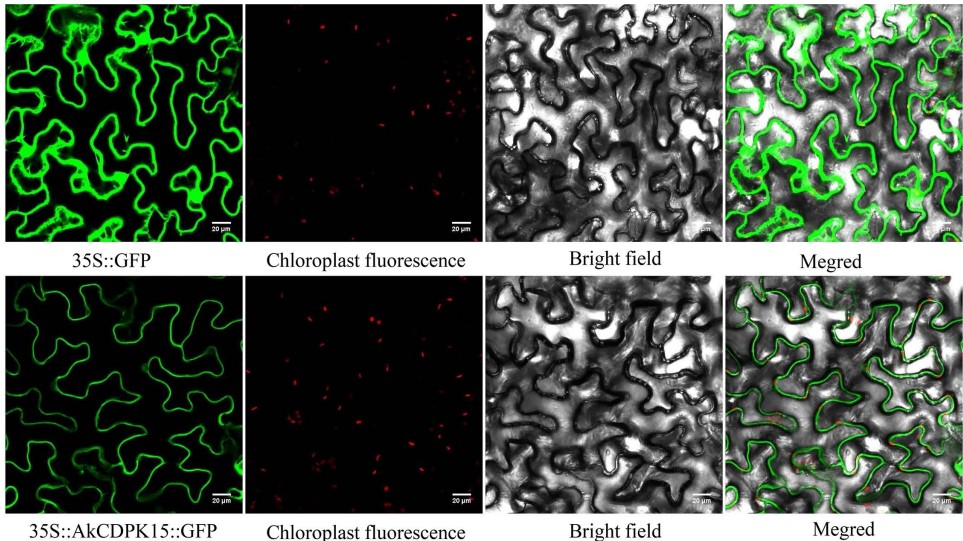

| 35S::GFP | Chloroplast fluorescence | Bright field | Megred |

| 35S::AkCDPK15::GFP | Chloroplast fluorescence | Bright field | Megred |

**Fig 9. Subcellular localisation of AkCDPK15.**

### *AkCDPK15* overexpression enhanced drought and salt tolerance

To understand *AkCDPK15* function under drought and salt conditions, we generated several overexpressing transgenic tobacco lines. Plants transformed with the empty vector were considered WT. Three *AkCDPK15*-overexpressing strains (OE1, OE2, and OE3) were used in further experiments. The root elongation experiments showed that the roots of the WT strain were significantly shorter than those of the *AkCDPK15*-overexpressing strains treated with 100 mM mannitol and 200 mM NaCl (Fig 10A, B, C, D). In contrast, there was no difference in the root length between the WT and untreated *AkCDPK15*-overexpressing strains (Fig 10E, F).

To further explore the function of *AkCDPK15*, one-month-old *AkCDPK15* transgenic and WT strains were subjected to water deficit stress. Under normal growth conditions, there were no significant differences in the growth of *AkCDPK15* transgenic and WT strains (Fig 11A). After 14 days of drought treatment, the degree of leaf wilting in the WT strain was more severe than that in *AkCDPK15*-overexpressing strains (Fig 11B). Two days after re-watering, both the WT and *AkCDPK15*-overexpressing strains showed growth recovery. However, yellowing and wilting of the bottom leaves of the WT strains were still higher than those of the *AkCDPK15*-overexpressing strains (Fig 11C). Taken together, *AkCDPK15*-overexpressing strains showed stronger resistance to drought stress than did the WT strains.

To understand the physiological changes in *AkCDPK15*-overexpressing and WT strains under drought stress, the MDA, proline, soluble sugar, and $H_2O_2$ contents, as well as the SOD, POD, and CAT activities of the two strains after 14 days of water deficit were detected. The results showed that after 14 days of drought stress, the MDA content of the WT strain was approximately 2.08 times higher than that of the *AkCDPK15*-overexpressing strain (Fig 11D). The proline and soluble sugar contents of the *AkCDPK15*-overexpressing strain were approximately 3.03 and 1.77 times those of the WT strain, respectively (Fig 11E, F). The $H_2O_2$ content of the WT strain was approximately 2.39 times that of the *AkCDPK15*-overexpressing strain (Fig 11G). However, the SOD, POD, and CAT activities of the *AkCDPK15*-overexpressing strain were 3.09, 2, and 1.63 times higher than those of the WT strain, respectively (Fig 11H, I, G). No significant differences in physiological indicators were detected under normal cultivation conditions (Fig 11). In short, overexpression of *AkCDPK15* in tobacco can enhance tobacco resistance to drought stress by reducing membrane damage in plant cells, enhancing antioxidant enzyme activity, and increasing intracellular stress resistance.

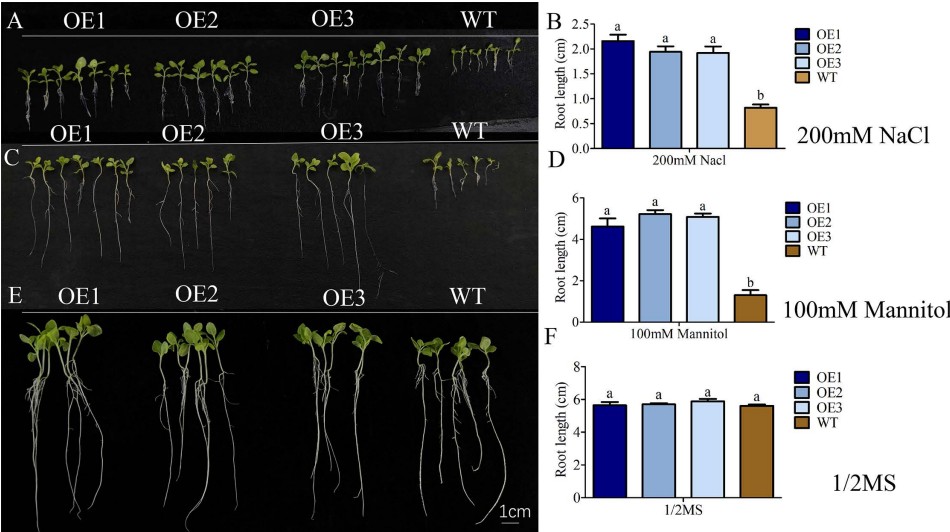

**Fig 10. Overexpression of *AkCDPK15* improves tolerance of tobacco to drought and salt stress. (A)** *AkCDPK15*-overexpressing and wild-type strains grown for 14 days on standard 1/2 Murashige and Skoog (MS) medium supplemented with 200 mM NaCl. **(B)** The root length bar chart of *AkCDPK15*-overexpressing and wild-type strains grown for 14 days on standard 1/2 MS medium supplemented with 200 mM NaCl. **(C)** *AkCDPK15*-overexpressing and wild-type strains grown for 14 days on standard 1/2 MS medium supplemented with 100 mM mannitol. **(D)** The root length bar chart of *AkCDPK15*-overexpressing and wild-type strains grown for 14 days on standard 1/2 MS medium supplemented with 100 mM mannitol. **(E)** *AkCDPK15*-overexpressing and wild-type strains grown for 14 days on standard 1/2 MS medium without supplementation. **(F)** The root length bar chart of *AkCDPK15* gene overexpression strains and wild-type strains grown for 14 days on standard 1/2 MS medium without supplementation. Different lowercase letters indicate significant differences, as calculated by Student's *t*-test).

## Discussion

Konjac corms are rich in glucosinolates and have broad application prospects in food, health, biomedicine, and industry, making them highly valuable crops. However, konjac is often affected by a variety of biotic and abiotic stressors, leading to a significant decline in its yield and quality. CDPK family genes not only exist in various plant tissues, such as roots, stems, and flowers, but also participate in various plant growth and development processes and respond to environmental stresses. In this study, a screening for *CDPK* family members from the *A. konjac* genome resulted in the identification of 29 members, divided into four subfamilies (Groups I-IV), together with an analysis of their structure, evolution, and expression profiles in response to phytohormones, and drought, salt, and Pcc stress.

Previous studies on evolutionary analysis of *CDPK* family members in *Arabidopsis* [37], rice [38,39], and patchouli [40] found that they are often divided into four subgroups. Similarly, the 29 *AkCDPK* family genes were divided into four subgroups. This indicates some degree of similarity in the evolution of the *CDPK* family members and in the diversity of gene structures among different species. In addition, we also found that *AkCDPK* family members contain 5–11 introns, similar to *Medicago truncatula* [41] and pineapple [42]. The number of introns among members of *AkCDPK* Group III in particular showed significant variability. Nuruzzaman et al. published that the rate of intron loss is higher than intron acquisition [43]. Therefore, we hypothesised that *AkCDPK* Group III may contain the original genes. Moreover, we noticed that in Group III, *AkCDPK* members contained a unique conserved motif 10, indicating the conservation of gene structural evolution in Group III.

The distribution of *CDPK* family members on plant chromosomes is not uniform. For example, in tomato they are mainly distributed on chromosomes 1, 10, and 11 [44], whereas in cucumber they are mainly distributed on chromosomes 2 and 5 [45]. In this study, *AkCDPK* family members were mainly distributed on chromosomes HIC_ASM_0, HIC_ASM_4,

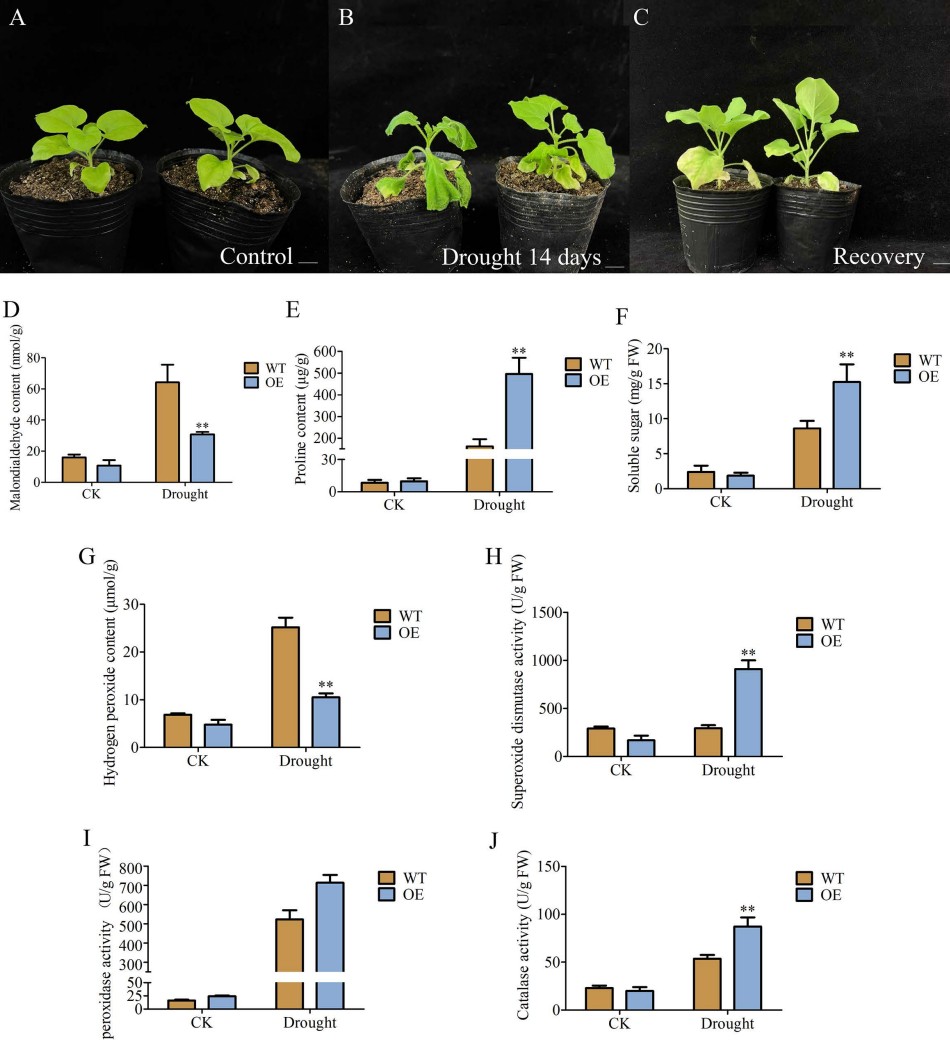

**Fig 11. Phenotypes and physiological indices of *AkCDPK15*-overexpressing tobacco strains under normal and drought stress conditions. (A)** One-month-old WT strains grown under normal conditions. **(B)** *AkCDPK15*-overexpressing and WT strains after 14 days without water. **(C)** *AkCDPK15*-overexpressing and WT strains in (B) after re-watering for two days. **(D-J)** The content of malondialdehyde (MDA) **(D)**, proline **(E)**, soluble sugar **(F)**, $H_2O_2$ **(G)** in WT and *AkCDPK15*-overexpressing strains under normal and drought stress. The activity of superoxide dismutase (SOD) **(H)**, peroxidase (POD) **(I)**, and catalase (CAT) **(J)** in WT and transgenic lines under normal and drought stress conditions. Data are presented as mean±SD (n=3, *$P < 0.05$; **$P < 0.01$, Student's *t*-test).

and HIC_ASM_5, but not on every chromosome. Gene replication events lead to fragment and tandem duplications, which play crucial roles in genome rearrangement and expansion [46]. In this study, there were only two collinear gene pairs within the family, indicating that *AkCDPK* had fewer copies or that copies were discarded more frequently during evolution. To further study the homologous relationships of *AkCDPK* genes in plants, an interspecific collinearity analysis was performed on *Arabidopsis* and rice. Two *A. konjac* genes (*AkCDPK11* and *AkCDPK21*) and two *Arabidopsis* genes were identified as syntenic orthologs between *A. konjac* and *Arabidopsis*, five *A. konjac* genes (*AkCDPK9*, *AkCDPK15*, *AkCDPK17*, *AkCDPK24*, and *AkCDPK29*), and seven rice genes were identified as syntenic orthologs between *A. konjac* and rice and these genes were distributed in four subgroups. We speculate these seven genes play crucial roles in the

evolution of the *AkCDPK* family. Genetic evolutionary selection analysis revealed that most *AkCDPK* were purified and selected, which was consistent with the results obtained for Chinese cabbage (*Brassica rapa*) [47] and *M. truncatula* [41].

There were multiple cis-acting elements in *AkCDPK* family member promoter regions that responded to plant hormones, stress, growth, and development, suggesting that the expression of *AkCDPK* may be regulated by phytohormones and stress, as well as being related to plant growth and development. This finding is consistent with the studies on *Arachis hypogaea* [48] and *Hevea brasiliensis* [42]. Moreover, RNA-Seq analysis of *AkCDPK* in different tissues revealed tissue specificity in *AkCDPK* expression. For example, *AkCDPK2*, *AkCDPK20*, and *AkCDPK17* were not expressed in the leaf, petiole, root and corm of *A. konjac*, and the *AkCDPK17* homologous genes *AtCPK17* and *AtCPK34* are involved in the development of *Arabidopsis* pollen tubes [49], whereas the *AoCPK16* gene (*AoCPK16* gene is homologous to *AtCPK17* and *AtCPK34*) has high expression levels in pineapple flowers and leaves [42]. *CDPK* is also involved in the response to various environmental stressors. For example, in *Arabidopsis*, under cold stress, the AtCPK28 phosphorylation cascade is activated to regulate transcriptional reprogramming downstream of $Ca^{2+}$ signalling, positively regulating the response to cold stress [50]. In wheat, the *TaCDPK27* gene enhanced resistance to salt stress by reducing ROS production, increasing antioxidant enzyme (POD, SOD, and CAT) activity, and reducing damage to photosystem II [51]. Similarly, in potatoes, *StCDPK2* actively participates in salt resistance by enhancing the antioxidant system [52]. In rape (*Brassica napus*), BnaCPK5 interacts with the ABA-responsive element-binding factors, BnaABF3 and BnaABF4, enhancing its drought resistance [53]. In rice, *OsCPK10* positively mediates plant tolerance to drought and rice blast disease by enhancing their antioxidant capacity and protecting them from ROS damage [54]. In this study, all 18 tested genes were induced by mannitol and salt, and 16 genes were induced by Pcc. The expression level of the Group I gene *AkCDPK15* was significantly altered under mannitol-, salt-, and Pcc-induced stress. Therefore, the ORF sequence of *AkCDPK15* was cloned and its response to drought and salt was analysed using tobacco *AkCDPK15*-overexpressing tobacco strains. Plants under drought stress often experience changes in a series of physiological indicators, such as cell membrane damage, accumulation of oxides, and changes in enzyme activity [55,56]. A dot plate experiment showed that the root length of *AkCDPK15*-overexpressing strains was significantly longer than that of the WT strain under mannitol (100 mM) and salt (200 mM NaCl) stress for 14 days. Furthermore, after 14 days water deficit, the activity of POD, SOD, and CAT and the content of proline and soluble sugar of the *AkCDPK15*-overexpressing strains were significantly higher than those of the WT strain, whereas the MDA and $H_2O_2$ contents were significantly decreased compared to those of the WT strains. Under normal cultivation conditions, there were no significant differences in these physiological indicators between the *AkCDPK15*-overexpressing and WT strains. Taken together, these findings show that *AkCDPK15* positively regulates drought and salt resistance in transgenic tobacco.

## Conclusion

In brief, 29 *AkCDPK* genes of *A. konjac* were identified, which could be divided into four subgroups and were unevenly distributed on 12 chromosomes. The gene motifs of these *AkCDPK* genes were highly conserved. There was one segmentally duplicated gene pair and one tandemly duplicated gene pair in the *AkCDPK* family. Between *A. konjac* and *Arabidopsis*, there were two syntenic blocks from CDPKs and six between *A. konjac* and rice. Furthermore, most *AkCDPK* genes underwent purifying selection during the evolutionary process. What's more, based on the predicted *AkCDPK* promoters, those *AkCDPK* genes are related to phytohormone induction, defence and stress responses, and other functions. *AkCDPK* exhibited tissue specificity, and the qRT-PCR analysis showing that *AkCDPK* responded to salt, drought, and Pcc stress. Finally, *AkCDPK15* was cloned, and shown to positively regulate plant drought and salt tolerance.

## Supporting information

**S1 Table. CDPK amino acid sequences of *A. thaliana*, *O. sativa*, and *A. konjac*.**
(XLSX)

**S2 Table. NCBI accession IDs for the transcriptome of *A. konjac.***
(XLSX)

**S3 Table. qRT-PCR primers.**
(XLSX)

**S4 Table. Duplications of *CDPK* genes among *A. konjac*, *A. thaliana,* and *O. sativa.***
(XLSX)

**S5 Table. Ka/Ks of the *AkCDPK* family.**
(XLS)

**S6 Table. Annotation of cis-acting regulatory elements in *AkCDPK* promoters.**
(XLSX)

**S7 Table. AkCDPK15 phosphorylation site information.**
(XLSX)

**S1 Fig. Western blot gel electrophoresis of *AkCDPK15.***
(TIF)

**S2 Fig. Gel electrophoresis of identification of *AkCDPK15* transgenic strains.**
(TIF)

**S3 Fig. Identification map of phosphorylation sites in AkCDPK15.**
(TIF)

## Author contributions

**Data curation:** Penghua Gao, Ying Zou, Lifang Li, Jianwei Guo, Yongteng Zhao, Jiani Liu.

**Funding acquisition:** Lifang Li, Ying Qi, Lei Yu.

**Methodology:** Ying Qi.

**Resources:** Jianrong Zhao.

**Validation:** Penghua Gao, Lei Yu.

**Writing – original draft:** Penghua Gao, Feiyan Huang, Lei Yu.

**Writing – review & editing:** Penghua Gao, Min Yang, Lifang Li, Jianwei Guo, Jiani Liu, Feiyan Huang, Lei Yu.

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
