## [Decision Letter · Decision Letter 0]

27 Feb 2025

Dear Dr. gao,

We look forward to receiving your revised manuscript.

Kind regards,

Sumita Acharjee

Academic Editor

PLOS ONE

“This study was funded by Yunnan Education Department Research Project (grant no. 2022J0644, 2023J0827), Applied Basic Research Foundation of Yunnan Province (grant no. 202301AU070136, 202301AT070055), Kunming University Talent Program (grant no. YJL23010, YJL23005, YJL23007), Yunnan Key Laboratory of Konjac Biology, Yunnan Province Yu Lei Expert Grassroots Research Workstation, Yunnan Provincial Science and Technology Department (grant no. 202201AU070043, 202101BA070001-174), Yunnan Provincial Science and Technology Department (grant no. 202449CE340009), College Student Innovation Training Program Project (grant no.S202411393051, S202411393042)”

Reviewers' comments:

Reviewer's Responses to Questions

**Comments to the Author**

1. Is the manuscript technically sound, and do the data support the conclusions?

Reviewer #1: Yes

Reviewer #2: Partly

2. Has the statistical analysis been performed appropriately and rigorously?

Reviewer #1: Yes

Reviewer #2: I Don't Know

3. Have the authors made all data underlying the findings in their manuscript fully available?

Reviewer #1: Yes

Reviewer #2: Yes

4. Is the manuscript presented in an intelligible fashion and written in standard English?

Reviewer #1: Yes

Reviewer #2: No

Reviewer #1: Review of Manuscript: Identification of the AKCDPK Gene Family and AkCDPK15 Functional Analysis under Drought and Salt Stress

The manuscript presents an interesting study on the AkCDPK gene family in Amorphophallus konjac, highlighting its evolutionary relationships and functional significance in stress tolerance.

The scientific content is relevant and well-structured, but there are language and clarity issues that should be improved.

Figures title and tables appear within the text, but there are formatting inconsistencies.

The figure resolution is poor, making it difficult to interpret the data.

Some sections have redundancies and unclear phrasing, which could be refined for clarity.

I give a few examples:

Lines 91-92: move it to the discussion

Lines: 98-99: it “Nine A. konjac seedlings” is it as 9 replicates? If yes, please replicates

Line: 100: why “two-month-old seedlings were used? Any reason? Why not one month?

lines 105-106: “The samples were collected at 0, 24, 48, and 72 h.” 0 should be converted to “samples were collected before teatment and 24, 48 and 72 h after treatment”.

Line 211: “First, transgenic (OE1, OE2, OE3)”, OE should be mentioned as full name when first use.

Lines 215- 216: “Next, one-month-old AkCDPK15-overexpressing and WT tobacco strains were subjected to water deficit treatment”. Make it more clear.

Line 232: “MW”, five full name when first use.

Line 242: Table 1: the table should be in landscape format in order to be able to read the heads. Especially column “GRAVY”

Line 247: NJ should be given as full name when first use.

Lines 252-255: should be moved from resutls to material and methods.

Lines 277-278: should be moved from resutls to material and methods.

Lines 282, 284: I think motif means AkCDPKa, if right, then covert motif 13 to AkCDPKa 13.

Line 408: “WT strain were significantly shorter than those”. Please give average value or percentage when say shorter than…..

Line 467: I recommend to use “abiotic stresses” than “abiotic stressors”

Reviewer #2: The manuscript by Gao et al. entitled as Identification of the AKCDPK gene family and AkCDPK15 functional analysis under drought and salt stress screened important CDPK family genes in the Konjac genome, studied their gene structure, evolutionary relationship, synteny-duplication and expression pattern. I have enjoyed reading the MS. This paper would be useful for Konjac improvement however current version is poorly written and rationale for the experiment is missing. I may recommend to accept after the following revision.

1. Mention the importance of Konjac, what are health, medicinal and economic benefits for growers and mankind. Why it is necessary to grown Konjac?

2. Manuscript title doesn’t mention Konjac plant name, the title need to revise. For example, “Identification of the AKCDPK gene family in Konjac and functional analysis of AkCDPK15 under drought”

3. What is the rationale to select 200 mM NaCl and 100 mM mannitol to induce salt and drought stress? How authors selected these concentrations?

4. Line 104; “An inoculation site was selected for each plant”- What does it mean?

5. Did authors include Konjac specific HMM model, which makes the analysis complete?

6. What is the basis of comparing phylogenetic relationship with rice? What are the other closely related tubers genome is available to for comparison and phylogeny?

7. Line 312-313, implying that these AkCDPK genes had experienced strong purifying selective pressure – What is the significance of this strong purifying selective pressure of CDPK genes?

8. Figure resolution is poor, please improve and re-submit.

**Do you want your identity to be public for this peer review?** For information about this choice, including consent withdrawal, please see our Privacy Policy

Reviewer #1: No

Reviewer #2: **Yes: ** Sushil S. Chhapekar

---

## [Author Response · Author response to Decision Letter 1]

9 Apr 2025

Reviewer #1: Review of Manuscript: Identification of the AKCDPK Gene Family and AkCDPK15 Functional Analysis under Drought and Salt Stress

The manuscript presents an interesting study on the AkCDPK gene family in Amorphophallus konjac, highlighting its evolutionary relationships and functional significance in stress tolerance.

The scientific content is relevant and well-structured, but there are language and clarity issues that should be improved.

Figures title and tables appear within the text, but there are formatting inconsistencies.

The figure resolution is poor, making it difficult to interpret the data.

Some sections have redundancies and unclear phrasing, which could be refined for clarity.

I give a few examples:

Lines 91-92: move it to the discussion

Response: Thank you for your suggestions, which we feel have improved our manuscript. We agree with your suggestion, so based on your advice, we have moved lines 91-92 to the discussion section.

Lines: 98-99: it “Nine A. konjac seedlings” is it as 9 replicates? If yes, please replicates

Response: We apologise for any confusion. In our experiment, each treatment was set up with three replicates, and each replicate contained nine A. konjac seedlings. This now reads “Each treatment was performed with three replicates, and each replicate contained nine A. konjac seedlings” (Lines 105–106).

Line: 100: why “two-month-old seedlings were used? Any reason? Why not one month?

Response: A. konjac sprouts relatively slowly, so using two-month-old seedlings ensures that the A. konjac leaves have fully expanded. Therefore, we selected using two-month-old A. konjac seedlings. We have included a brief mention of this rationale (Lines 110–111).

lines 105-106: “The samples were collected at 0, 24, 48, and 72 h.” 0 should be converted to “samples were collected before teatment and 24, 48 and 72 h after treatment”.

Response: We have made revisions based on your suggestion. The revised section has been highlighted in red font. The new text reads “Samples were collected before treatment and 24, 48 and 72 h after treatment” (Lines 115–116).

Line 211: “First, transgenic (OE1, OE2, OE3)”, OE should be mentioned as full name when first use.

Response: We have made revisions based on your suggestion. The revised section has been highlighted in red font This now reads “(AkCDPK15 overexpression strain 1 [OE1], AkCDPK15 overexpression strain 2 [OE2], AkCDPK15 overexpression strain 3 [OE3])” (Lines 228–229).

Lines 215- 216: “Next, one-month-old AkCDPK15-overexpressing and WT tobacco strains were subjected to water deficit treatment”. Make it more clear.

Response: We have made revisions based on your suggestions. The revised sections have been highlighted in red font.

Line 232: “MW”, five full name when first use.

Response: We have revised this accordingly (Line 252).

Line 242: Table 1: the table should be in landscape format in order to be able to read the heads. Especially column “GRAVY”

Response: We have altered the table to make it easier to read.

Line 247: NJ should be given as full name when first use.

Response: We have revised this based on your suggestion. The revision has been highlighted in red.

Lines 252-255: should be moved from resutls to material and methods.Lines 277-278: should be moved from resutls to material and methods.

Response: We have moved these portions based on your suggestions.

Lines 282, 284: I think motif means AkCDPKa, if right, then covert motif 13 to AkCDPKa 13.

Response: In the manuscript, 'motif' refers to the conserved motifs contained within the genes of various CDPK family members; therefore, we would like to maintain this as it is.

Line 408: “WT strain were significantly shorter than those”. Please give average value or percentage when say shorter than…..

Response: We have made revisions based on your suggestions. The revised section has been highlighted in red font. This now reads “In terms of salt stress (200 mM NaCl), the WT strain was substantially shorter than the transgenic strain, with an average height of 0.92 cm compared to 2.13 cm (Fig 10A, B). In terms of mannitol stress (100 mM mannitol), the WT strain was substantially shorter than the transgenic strain, with an average height of 1.86 cm compared to 5.23 cm (Fig 10C, D)” (Lines 436–441).

Line 467: I recommend to use “abiotic stresses” than “abiotic stressors”

Response: Thank you. We have changed this accordingly.

Reviewer #2: The manuscript by Gao et al. entitled as Identification of the AKCDPK gene family and AkCDPK15 functional analysis under drought and salt stress screened important CDPK family genes in the Konjac genome, studied their gene structure, evolutionary relationship, synteny-duplication and expression pattern. I have enjoyed reading the MS. This paper would be useful for Konjac improvement however current version is poorly written and rationale for the experiment is missing. I may recommend to accept after the following revision.

1. Mention the importance of Konjac, what are health, medicinal and economic benefits for growers and mankind. Why it is necessary to grown Konjac?

Response: We would like to offer our gratitude for reviewing our manuscript. We have made additions in the introduction section, which have been highlighted in red font. The text now reads “The konjac corm is rich in glucomannan, a soluble dietary fibre that has cholesterol-lowering, blood sugar-regulating, detoxifying, and gut health-supporting properties. In addition, konjac also possesses antibacterial and anti-inflammatory effects [19, 20]. Based on these characteristics, the application of konjac in the biopharmaceutical, industrial, and food processing industries is becoming increasingly widespread [21]. As a result, the cultivation area of konjac continues to expand. It is also worth noting that konjac cultivation requires relatively low investment, providing farmers with opportunities to increase their income [22]” (Lines 76–83).

2. Manuscript title doesn’t mention Konjac plant name, the title need to revise. For example, “Identification of the AKCDPK gene family in Konjac and functional analysis of AkCDPK15 under drought”

Response: We have revised the title based on your advice.

3. What is the rationale to select 200 mM NaCl and 100 mM mannitol to induce salt and drought stress? How authors selected these concentrations?

Response: We have two reasons for choosing 200 mM NaCl and 100 mM mannitol.

First, we chose these values based on the results of preliminary experiments: During the testing of different concentrations, we found that 200 mM NaCl and 100 mM mannitol induced the most substantial stress phenotypes in plants. Specifically, we tested NaCl concentrations of 100, 150, and 200 mM, and mannitol concentrations of 50 and 100 mM.

Seconed, when reviewing the relevant literature, we found that similar concentrations were used in comparable studies. For example, Yu et al. in their study on rice WRKY gene salt tolerance (doi: 10.1038/s41467-023-39167-0), and Sattar et al. in their study on potato drought stress (doi: 10.3390/plants10050924) both used similar concentrations.

4. Line 104; “An inoculation site was selected for each plant”- What does it mean?

Response: “An inoculation site was selected for each plant” means “During PCC inoculation, each plant has only one inoculation site”. The revised sections have been highlighted in red font.

5. Did authors include Konjac specific HMM model, which makes the analysis complete?

Response: We apologise for any confusion. We have made the necessary revisions in our manuscript. The revised sections have been highlighted in red font.

6. What is the basis of comparing phylogenetic relationship with rice? What are the other closely related tubers genome is available to for comparison and phylogeny?

Response: Since both konjac is a monocotyledonous plant and rice is another widely studied monocotyledonous model plant, we selected rice to compare the phylogenetic relationship. Moreover, since the whole genomes of closely related tuber plants within the Amorphophallus genus have not been published, we did not select other tuber plant genomes for comparison and phylogenetic analysis.

7. Line 312-313, implying that these AkCDPK genes had experienced strong purifying selective pressure – What is the significance of this strong purifying selective pressure of CDPK genes?

Response: We have revised this in our manuscript. The revised sections have been highlighted in red font.

8. Figure resolution is poor, please improve and re-submit.

Response: We have re-submitted these figures.

And in the last, our manuscript has undergone language editing and refinement by Editage. This is our editing certificate.

---

## [Decision Letter · Decision Letter 1]

13 May 2025

Identification of the AKCDPK gene family and AkCDPK15 functional analysis under drought and salt stress

PONE-D-24-59338R1

Dear Dr. gao,

We’re pleased to inform you that your manuscript has been judged scientifically suitable for publication and will be formally accepted for publication once it meets all outstanding technical requirements.

Kind regards,

Sumita Acharjee

Academic Editor

PLOS ONE

Additional Editor Comments (optional):

Reviewers' comments:

Reviewer's Responses to Questions

**Comments to the Author**

Reviewer #2: All comments have been addressed

2. Is the manuscript technically sound, and do the data support the conclusions?

Reviewer #2: Yes

3. Has the statistical analysis been performed appropriately and rigorously?

Reviewer #2: Yes

4. Have the authors made all data underlying the findings in their manuscript fully available?

Reviewer #2: Yes

5. Is the manuscript presented in an intelligible fashion and written in standard English?

Reviewer #2: Yes

Reviewer #2: Author have addressed my comments and I find lots of improvement in the manuscript.

The revised manuscript will be helpful for Konjac research.

**Do you want your identity to be public for this peer review?** For information about this choice, including consent withdrawal, please see our Privacy Policy

Reviewer #2: **Yes: ** Sushil Chhapekar

---

## [Editor Report · Acceptance letter]

PONE-D-24-59338R1

PLOS ONE

Dear Dr. Gao,

I'm pleased to inform you that your manuscript has been deemed suitable for publication in PLOS ONE. Congratulations! Your manuscript is now being handed over to our production team.

Kind regards,

on behalf of

Dr. Sumita Acharjee

Academic Editor

PLOS ONE